# Predicting Terrorism in Europe with Remote Sensing, Spatial Statistics, and Machine Learning

Caleb Buffa [1,2,3,]*, Vasit Sagan [1,2], Gregory Brunner [1,3] and Zachary Phillips [1,2]

1 Geospatial Institute, Saint Louis University, St. Louis, MO 63108, USA; vasit.sagan@slu.edu (V.S.); gbrunner@esri.com (G.B.); zachary.phillips@slu.edu (Z.P.)
2 Department of Earth and Atmospheric Sciences, Saint Louis University, St. Louis, MO 63108, USA
3 Environmental Systems Research Institute, 380 New York Street, Redlands, CA 92373, USA
* Correspondence: cbuffa@esri.com

**Abstract:** This study predicts the presence or absence of terrorism in Europe on a previously unexplored spatial scale. Dependent variables consist of satellite imagery and socio-environmental data. Five machine learning models were evaluated over the following binary classification problem: the presence or absence of historical attacks within hexagonal-grid cells of 25 square kilometers. Four spatial statistics were conducted to assess the validity of the results and improve our inferential understanding of spatial processes among terror attacks. This analysis resulted in a Random Forest model that achieves 0.99 accuracy in predicting the presence or absence of terrorism at a spatial resolution of approximately 5 km. The results were validated by robust F1 and average precision scores of 0.96 and 0.97, respectively. Additionally, statistical analysis revealed spatial differences between separatists and all other terrorist types. This work concludes that remote sensing, machine learning, and spatial techniques are important and valuable methods for providing insight into terrorist activity and behavior.

**Keywords:** machine-learning; remote sensing; spatial statistics; terrorism

## 1. Introduction

A precise definition of what terrorism is and what it is not the subject of debate dating back to the first use of the word in 1789, resulting in more than 109 legal definitions currently in use [1–3]. Due to the lack of clarity, this research has opted for a broad description of terrorism that can be understood as non-state actors' use of violence against noncombatants with a primary goal of political coercion [4]. This classification steers away from other definitions that may include state actors or threats of violence, as such definitions that may introduce ambiguity in data screening and the discussion that follows.

Bahgat and Medina, in a 2013 meta-analysis of counter terrorism studies, noted a fixation on political and sociological perspectives to the determent of geographic ones, primarily due to a lack of high-quality geo-referenced data sets and untapped applications of Geographic Information Systems [5]. However, terrorism exhibits an important spatial dimension. It stands to reason that if governmental policies are designated within a defined territory, terrorism can, in some part, be understood as an attempt to influence control over political boundaries [6]. Similarly, strong links between a lack of territorial security, and the presence of terrorism have been found [7–9], as a lack of governmental control allows terrorists to view the environment as permissive to their acquisition of an autonomous zone [10].

Furthermore, high-population areas and those of governmental importance are both strategically attractive and cost-effective targets, and cities with high global and regional regard increase their attractiveness to terrorist target selection [11]. Terrorists' specific motivations also have a geographic component, exemplified by religious groups targeting

civilians seen as practicing immoral behavior and places of worship [12]; right-wing terrorists selecting governmental buildings [13]; persecuted individuals seeking out civilians in everyday locations such as malls [14]; territorial terrorists operating in peripheral environments [15]. There is further a self-reinforcing cycle in which areas previously attacked become increasingly attractive to future attackers [16], and a relationship between terrorists' bases of planning and preparation [17–19] suggests target locations are a function of geographic convenience as much as an act of coercion.

Singh et al. combined Hidden Markov Chains to model, in real-time, activities deemed consistent with historical attacks and determine the best actions to prevent them [20]. Additionally, Dixon et al. developed a feed-forward neural network (NN) to identify deceptive behavior, producing an average accuracy of 0.6 with the best success rate of 0.68 [21]. An ensemble was also trained to predict the attributes of attacks with accuracy scores of 0.79 to 0.85 [22]. Mo et al. built a Support Vector Machine (SVM), Naïve Bayes classifier, and a Logistic Regression to predict terrorist events with a maximum accuracy of 0.78 [23].

Dixon et al. [24] developed a NN, SVM, and a Random Forest, in which the authors concluded 0.96 accuracy scores and robust classification reports in predicting attack locations, but noted the global aggregation reduced the model's sensitivity to regional and local factors.

Most recently, Uddin et al. [25] developed five machine learning models based on NNs to predict the outcomes of attacks, such as attack type and weapon used. To supplement NN architectures, the researchers employed logistic regression, SVM, and Naïve Bayes classification. A comparison of results concluded that DNNs are more successful than other algorithms, achieving 0.95 accuracies compared to 0.83 for non-DNN models. In doing so, the team concluded the successful creation of a model that accurately predicts the future lethality of a terror organization based solely on the first 10 attacks attributed to it.

While the research to date has reported robust results, little has been performed to understand terrorism at the sub-state level. It is well understood that the causes of terrorism at general levels and there are qualitative inferences as to why it occurs within specific regions, but what is less clear is whether terrorism can be predicted at smaller scales. Additionally, few studies have focused on remote sensing and spatial sciences.

This research aims to predict terrorism in Europe at the sub-national level. The objectives of the study are the following: (1) develop a model to predict terrorism using satellite imagery and socio-environmental data; (2) develop a grid-cell-based spatial statistics approach to reveal specific causal variables and trends in this region at a previously unexplored spatial scale.

## 2. Methods

This study includes 18,741 attacks occurring between 1970 and 2018 in Eastern and Western Europe apart from Russia. The target was developed using the National Consortium for the Study of Terrorism and Responses to Global Terrorism Database (GTD) [26]. The GTD is an open-source dataset of terrorist incidents since 1970 that has been georeferenced, matches the definitional criteria of the study, and has been the primary source of terrorism studies in the United States.

Remotely sensed data was collected from a variety of sources including the 2000 Shuttle Radar Topography Mission, 2020 Copernicus land cover [27], and 2018 nighttime lights from the visible and infrared imaging suite (VIIRS) sensor on board the Joint Polar-orbiting Satellite System [28]. Figure 1 visualizes the study and samples the remote sensing data used in the study.

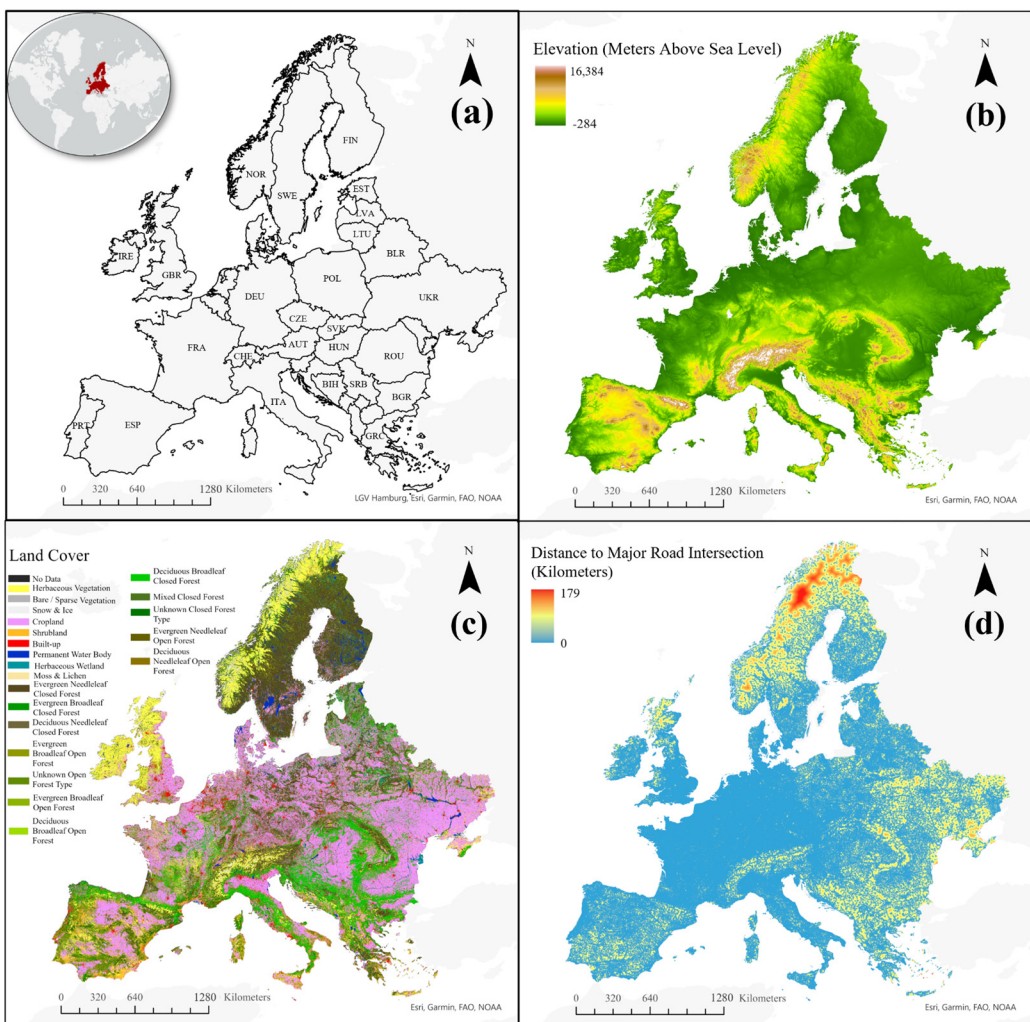

**Figure 1.** The study area (**a**) with sample explanatory variables. Attacks occurring between 1970 and 2018 within this geographic region are considered. Explanatory variables were collected from remotely sensed data including (**b**) digital elevation model, (**c**) land cover classification [8], and (**d**) distance to major road intersections. DEM and distance to major road intersections data can be found at www.worldpop.org (accessed on 10 October 2021).

Geospatial and population features were collected from WorldPop, a peer-reviewed research data archive. From this archive, data was obtained on distances to major waterways, inland water, major road intersections, roadways, population counts, population densities, built settlement growth, and demographics from 2000–2020. Civil unrest was calculated from the Armed Conflict Location and Event Dataset (ACLED) [29]. It should be noted as a limitation, however, that this study covers terrorism until 2018 (the last year GTD has data available at the time of the study) while civil unrest is calculated from data beginning in 2018 (the earliest year ACLED has data available at the time of the study). It is therefore assumed this feature is representative of larger civil unrest during the study's time period. For spatial-temporal analysis, the Integrated Crisis Early Warning System (ICEWS) [30], which documents socio-political interactions extracted via open-source documents, is used.

Shapefile boundaries were imported from the Department of State's International Boundaries data set. ACLED data was imported as vector points before being converted to raster format at a 10 km ground sampling distance (GSD). Pixel values were calculated as the number of events occurring within a pixel. Next, 5-square-kilometer hexagonal cells were created within the area of study and vector points placed at the centroid of each polygon. Points falling within a 3-km buffer of attack locations were removed. The spatial

resolution for each observation is therefore approximately 5-km and the results can be interpreted as such.

The dataset was imported into a Python 3.7 environment for processing. The 23 Copernicus land cover classifications were aggregated into urban, vegetation, agriculture, and water. The data was then randomly split into training (0.7), evaluation (0.15), and hold-out test splits (0.15). Evaluation data was used for comparison and hyper-parameter tuning while the hold-out test set was used only for final performance assessments, from which the reported results are derived.

An unpruned Random Forest of 100 trees was fit to the training data and permutation importance used to retrieve feature importance scores from the evaluation set to understand each feature's generalizability. This methodology performs robustly on random values, data sets containing both categorical and binary features, as well as in the presence of co-linearity [31].

Features with lower feature importance and correlation coefficients higher than 0.7 were removed apart from binary and categorical features. Lastly, the data was normalized to a 0–1 scale. This resulted in the following features used for prediction: (1) distance to inland waterway, (2) distance to major road, (3) distance to major waterway, (4) elevation, (5) civil unrest, (6) population density, (7) slope, (8) nighttime lights, as well as (9) urban and (10) agricultural land-cover. The study workflow is visualized in Figure 2.

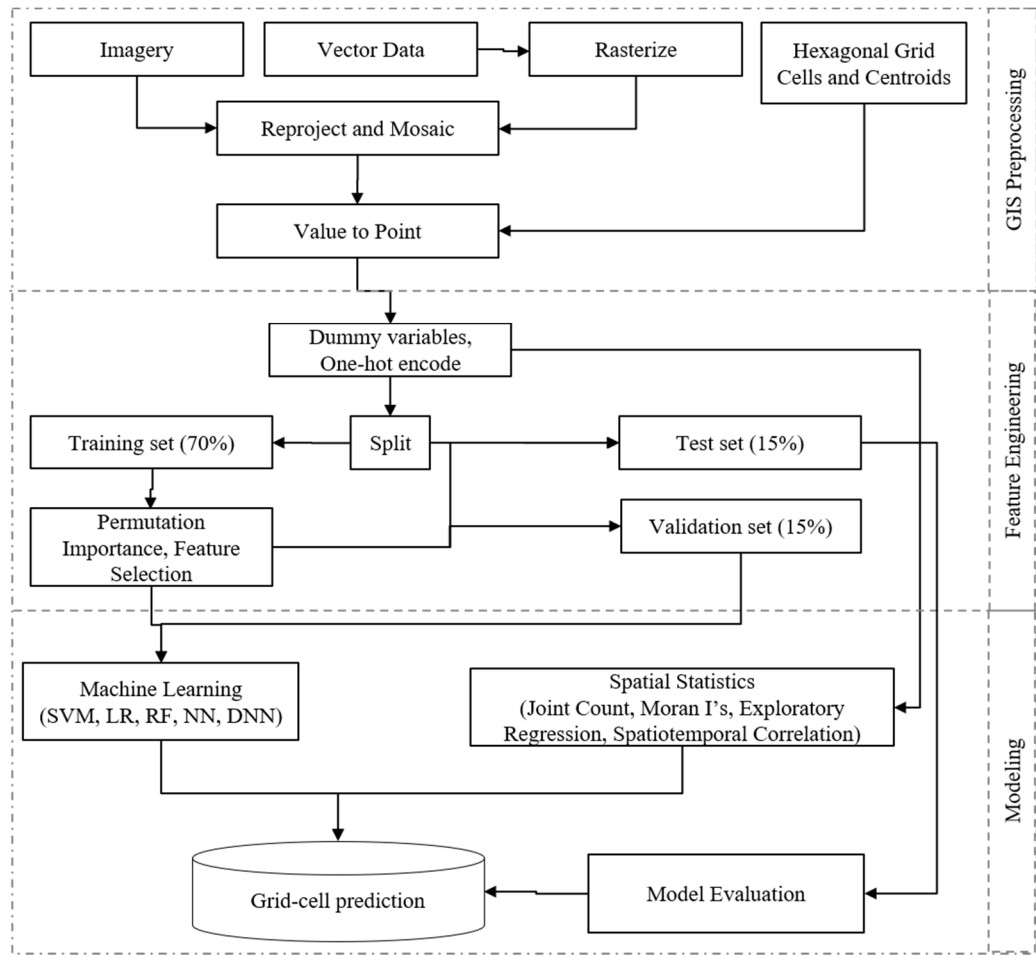

**Figure 2.** The methodology workflow. Cell values were sampled from processed raster and vector data via hexagonal grid centroids. Feature engineering was performed on train, validation, and test splits. Five machine learning models compared, and four spatial statistics methods used to output a final prediction map.

The predictor variables (Table 1) were chosen due to the interaction effect each likely has on the target variable. Proximity to major roadways provides convenient transportation to and from the target location while waterways may provide an additional, alternative, or backup means of escape for well-planned attacks in addition to the well-known geographic link between cities and bodies of water. Additionally, nighttime lights, population density, and land cover each track an important aspect of human geography. Naturally, terrorists are likely to attack urban centers with large numbers of civilians, and most likely, those that are well-developed. All of which is captured in the three variables. Slope and elevation were included to capture any geographic and spatial logic consistent among densely populated urban areas not previously captured; while civil unrest attempts to assess the public sentiment that may drive a terrorist to select, or not select, a specific location.

**Table 1.** Description of input features, the year they were collected, and original raster resolution. All variables, accounting for various spatial resolutions, were aggregated and sampled via hexagonal grid-cells.

| Feature | Description | Year | Resolution |
|---|---|---|---|
| Distance to inland water | Kilometers to inland body of water | 2016 | 100 m |
| Distance to major road | Kilometers to OSM roadway | 2016 | 100 m |
| Distance to major waterway | Kilometers to major navigable waterway | 2016 | 100 m |
| Elevation | SRTM meters above sea level | 2000 | 100 m |
| Civil unrest | Armed conflict location and event dataset | 2018 | 10 km |
| Population density | People per pixel | 2018 | 1 km |
| Slope | SRTM degree of topographic slope | 2000 | 100 m |
| Nighttime lights | VIIRS temporally calibrated nighttime lights | 2018 | 100 m |
| Landcover | Copernicus calibrated nighttime lights | 2018 | 100 m |

To evaluate model predictions, accuracy, average precision (AP), and F-1 scores are reported. However, due to accuracy's sensitivity to class imbalances [32], AP and F-1 are used for final assessment and comparison. Average Precision, $\sum n \ (R_n - R_{n-1}) \ P_n$ , is informative of the precision-recall curve, with $R_n$ and $P_n$ being the precision and recall scores at the $n$th step, while F-1, $\frac{2 * (P*R)}{P+R}$, is the harmonic mean of precision and recall globally [33]. Both metrics are therefore more robust and overcome overly optimistic metrics by assessing Type 1 and Type 2 errors [34].

Logistic regression classifies input data by maximizing the log odds that an observation belongs to the $n$th class. In doing so, it provides intuitive coefficients and class probabilities allowing for excellent inferential interpretation. Additionally, it has been shown to perform well in the face of more flexible non-parametric models such as Support Vector Machines and Neural Networks [35,36]. This model was optimized with stochastic gradient descent [37], leveraging an alpha value of $1 \times 10^{-6}$, and an initial learning rate of 0.1, producing an 0.87 F-1 score.

The Random Forest classifier, by aggregating the predictions of multiple weak decision trees, has found success among co-linear features, provides fast convergence, is largely unresponsive to over-fitting [38], and provides impressive performance in handling categorical and continuous variables, unbalanced data, outliers, and in a variety of fields

including remote sensing [39]. This study's data set, containing categorical and continuous variables, highly unbalanced features, and by nature, all instances belonging to the positive class are outliers, suggests Random Forest is an ideal model for the topic. This model leveraged a max depth of 3 trees, Gini impurity, and considered a maximum of 3 features for each split, producing an F-1 of 0.96.

Support Vector Machines, well known for the "kernel trick" [40], delineate non-linearly separable features by mapping data points into an *n*-dimensional kernel to avoid the computational cost of calculating *n*-dot products in a 3-D feature (Equation (1)). The primary parameters of the Radial Basis Function kernel, gamma, and C were sampled from logarithmic grids of ranges $1.0 \times 10^{-4}$–$1.0 \times 10^4$ and $1.0 \times 10^{-2}$–$1.0 \times 10^{10}$, respectively. However, despite the kernel trick's success, SVMs are computationally expensive as fit times increase quadratically with the number of observations [41]. As a result, ten SVMs were trained in a bagged ensemble. Each child estimator was exposed to a random sample of 0.1 of the training data without replacement, decreasing fit times and improving generalization [42]. This model, utilizing the RBF kernel, leveraged a C and gamma of 1.0 and $6.9 \times 10^{-3}$, resulting in a 0.87 F-1.

$$K(x, \ x_i) = exp\left(\frac{|x - x^i|}{2\sigma^2}\right) \tag{1}$$

Neural networks, comprise an arbitrary number of layers connected by an arbitrary number of neurons, allowing it to learn non-linear decision boundaries via non-linear activation functions. The size of the model, both in terms of layers and neurons per layer, impacts the level of complexity that can be learned. However, larger models will often degrade in performance past a consistent depth due to the number of matrix computations performed. Therefore, it cannot be stated that a deeper or larger model will necessarily perform better. Only that, with proper techniques to avoid vanishing/exploding gradients and over-fitting, it will not likely perform significantly worse.

The shallow NN, obtained from Ref. [25], consisted of a single hidden layer of 10 neurons with an initial learning rate of $1 \times 10^{-3}$, 500 epochs of training, Adam optimization, logistic activation function, and cross entropy loss. The DNN (Figure 3) leveraged the same hyperparameters as the single layer-NN, but with 5 layers consisting of 100, 50, 30, 10, and 5 neurons. The neural networks achieved a 0.91 and 0.92 F-1 score, respectively.

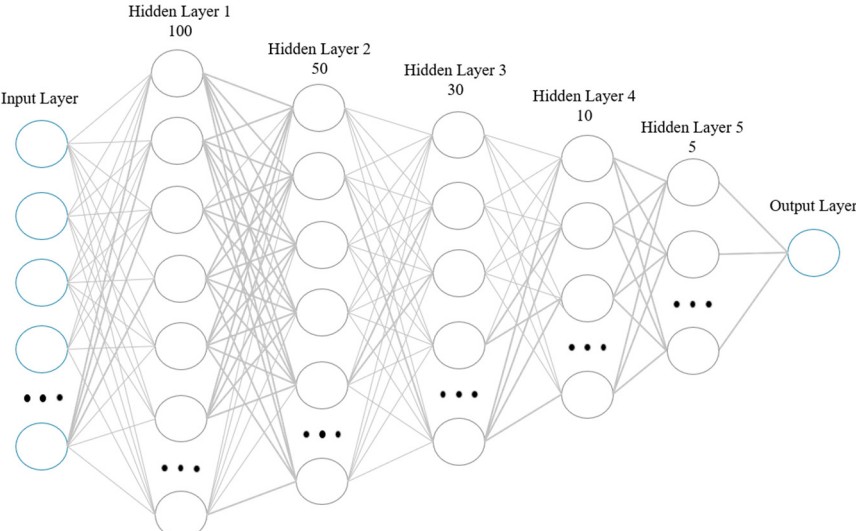

**Figure 3.** Uddin et al. DNN architecture [25] used in the study outperformed larger and more complex models with regularization built for comparison producing an F1 score of 0.92.

## 3. Results

Table 2 compares the results across accuracy, average precision computed from precision-recall curves, and F1 scores. Average precision and F1 were chosen as the evaluation metrics due to their robustness in the face of large class imbalances. Each model performed exceptionally well, but the Random Forest outperformed every other across all tasks, with Figure 4 visualizing feature importance scores obtained from the Random Forest model.

**Table 2.** Classification report results. Random Forest achieved the highest F1 and average precision scores. Due to class imbalances, F1 and AP are considered the primary evaluation metrics. Random Forest is therefore used for final predictions.

| Model | Accuracy | AP | F1 |
|---|---|---|---|
| DNN | 0.98 | 0.91 | 0.91 |
| NN | 0.98 | 0.90 | 0.91 |
| Random Forest | 0.99 | 0.97 | 0.96 |
| Log Reg + SGD | 0.96 | 0.90 | 0.88 |
| SVM Ensemble | 0.96 | 0.63 | 0.88 |

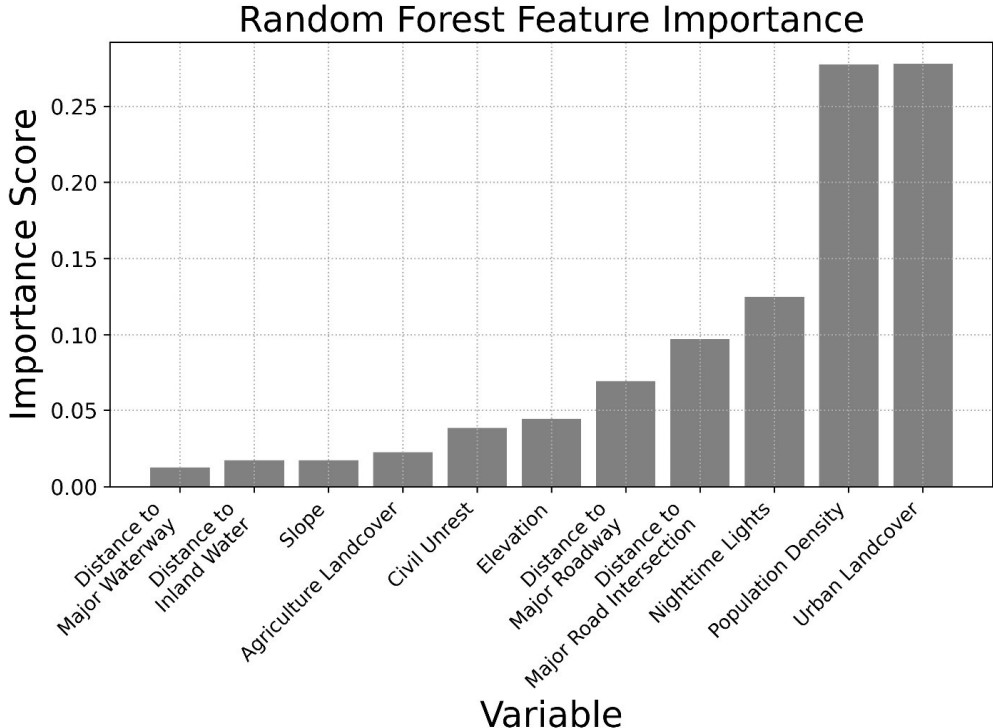

**Figure 4.** Random Forest feature importance scores. Urban landcover classification, population density, and nighttime lights are the most important features impacting model prediction. These findings are in line with previous qualitative studies and enhance our understanding of terrorist target selection.

Spatial statistics were then used to validate the Random Forest's spatial accuracy. Join counts were used to assess the spatial correlation between the classifier's predictions. We can understand spatial processes and distributions as one of many, or potentially infinite, possibilities. Therefore, by modeling random distributions, join counts compare synthetic results with the observed values to assess the likelihood of geographic correlation.

The Random Forest did not demonstrate a statistically significant correlation among incorrect predictions at large, with a synthetic *p*-value of 0.191. With a high simulated *p*-value, this research can reject the null hypothesis that the results were clustered, and thus, spatial correlation is not occurring among incorrect predictions at large. However,

the model did exhibit a significant spatial correlation in terms of false negative predictions, with a synthetic *p*-value of 0.001.

A DBSCAN model identified clusters of false negatives in the Donetsk region of Ukraine, Northern Ireland, Kosovo, the Basque country of Spain, and the island of Corsica, France (Figure 5).

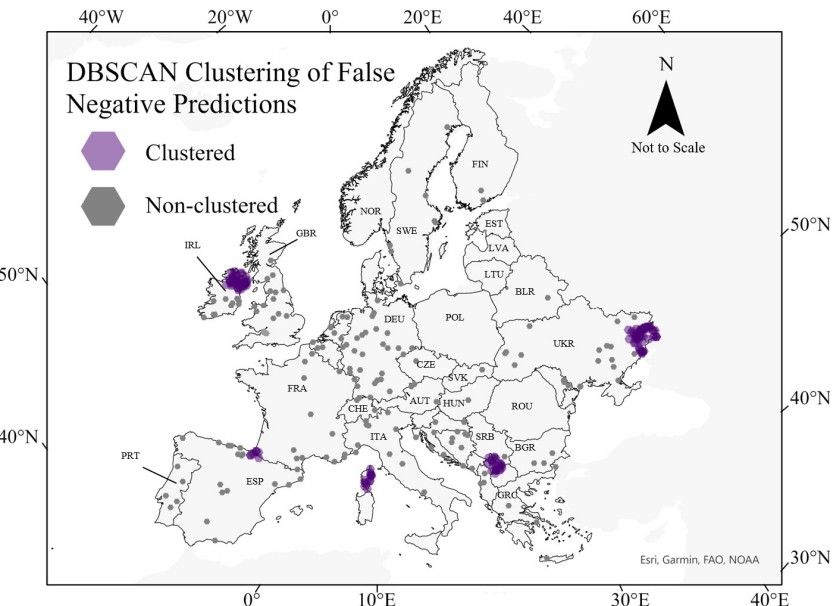

**Figure 5.** DBSCAN false negative clusters (purple). Clustering is occurring within Northern Ireland, Northern Spain, Eastern Ukraine, Kosovo, and Corsica, France. Each region has the following attribute in common: large-scale and ongoing separatist terrorism. Thus, suggesting a common spatial attribute among territorial terrorism separate from non-territorial. Note, clusters were enlarged for visualization purposes and are therefore not to scale. The map is in the Albers equal-area conic projection system.

## 4. Discussion

While the best-performing model, Random Forest, demonstrated clustered false negatives, this revealed an underlying trend. Each cluster is a known hot spot of terrorism in which territorial control is the primary grievance or objective. As noted earlier, Northern Ireland was inundated with ethno-nationalist conflict from the 1960s to the late 1990s [43]; the Basque country of Spain also saw armed conflict from the 1960s to the early 2000s between Basque organizations who sought independence [44]; Kosovo underwent upheaval with respect to autonomy in the 1990's between Yugoslavian and Kosovo nationalists [45]; Corsican nationalists have fought French, Italian, and Spanish security forces regarding independence from France [46]; most recently, conflict in Eastern Ukraine, stemming from the 2014 Russian invasion, has resulted in Russian separatists fighting for annexation into Russia [47].

While non-territorial terrorists target highly populated urban areas with symbols of their grievance, separatism is a function of operating on the periphery, where the current government does not retain high degrees of control. These locations are likely to occur outside of the most populated areas that are typical of other types of terrorism. This is likely the reason for clustering but validates the geographic differences between these types of terrorism. Furthermore, the lack of clustering among non-territorial terrorism suggests sub-classes of terrorist types (right-wing, left-wing, environmental, etc.) maintain similar geographical logic and planning. Thus, it may be more appropriate for future spatial studies to examine terrorism in terms of its territorial or non-territorial nature rather than the specific issues that drive them.

These findings can play an important role in counter-terrorism studies. First, given the Random Forest's outperformance of more complex models, it can be inferred that the complexity of terrorist target selection is not high, as can be deduced from an ensemble of decision trees. In fact, these results are such that it may be inferred that areas analysts believe to be likely targets based on common intuition and previously established literature are, in fact, the most likely targets, and rarely do terrorists venture outside of this. Additionally, the lack of significant relationships between ICEWS and GTD data suggests that terrorists are not likely to be mobilized by the larger socio-political discourse. It can be inferred that terrorists are either uninterested in public discourse or are largely disassociated from it. If the former is true, terrorism may be a marketing campaign of sorts, aimed at moving the direction of conversation towards an area of their favor. If the latter holds true, then more research should be performed to collect the interactions terrorists are interested in.

Final predictions (Figure 6) show the model correctly capturing current and past hot spots while removing some grids in which attacks have occurred and predicting new locations where they have not. For deployment in real-world applications, it is envisioned that points of interest would be sampled, and the results interpreted as a 25-square-kilometer area around the point. In this way, the results would prove more dynamic than the static map represented here. However, subject matter expertise in regional studies, terrorism, and machine learning should be considered when interpreting any results from the model, as terrorism is a complex topic with many caveats that no model can capture in its entirety, including, for example, the intricacies of lone wolf attacks.

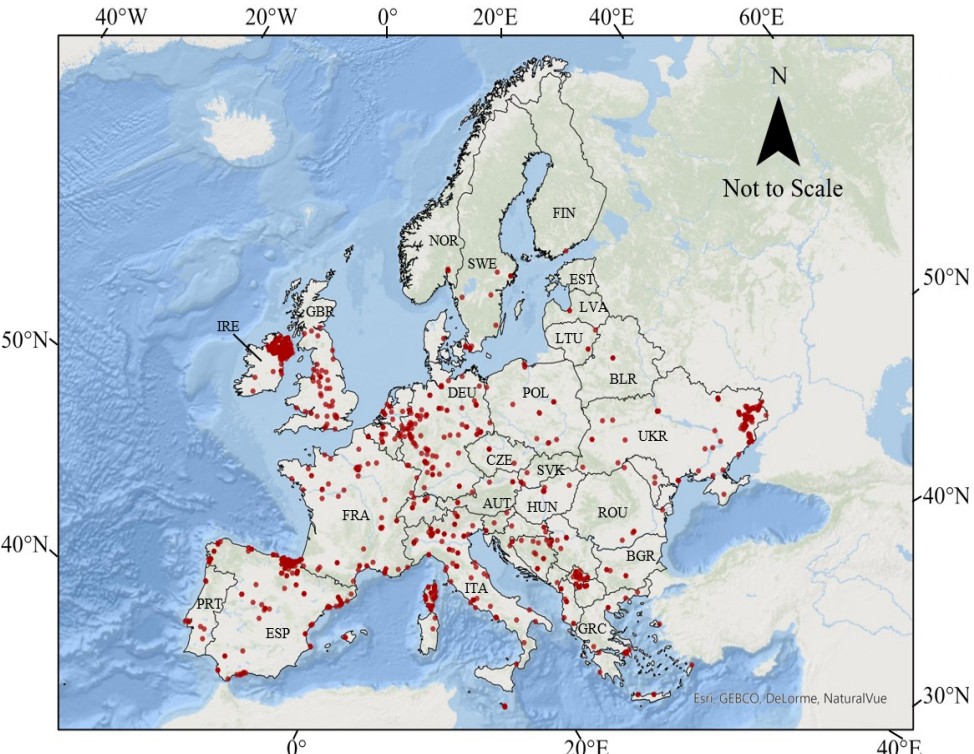

**Figure 6.** Random Forest predictions of terrorist target locations. Each red dot represents a grid-cell the model has predicted as consistent with historical targets for terrorism. Those grids that have been attacked before are at elevated risk, while those that have never been attacked are predicted to be at risk should the political and social environment lend itself. Please note grid cells have been enlarged for visualization and are therefore not to scale. The map is in the Albers equal-area conic projection system.

## 5. Conclusions

Although this study demonstrated robust results, there remain challenges in the field and limitations to the study. First, this study created a 25 sq. km hexagonal grid cell approach to understand where terrorism is likely to occur. However, it did not consider neighboring pixel values in the rasters from which the features were sampled. It may be deduced that if an attack occurred in an area deemed agricultural but on the edge of the pixel nearest urban areas, the attack location was not chosen due to its agricultural nature but its proximity to urban centers. However, attempts to include this spatial attribute degraded model performance and limited inference. Additionally, categorical variables could not be computed in this way as it would result in the loss of the land cover class entirely and introduce a continuous component to a categorical variable.

To ensure this limitation was not detrimental to the results of the study, multiple experiments were conducted at different grid-cell sizes (100 and 50 square kilometers), shapes (square), and random sampling. Across all experiments, results differed marginally, by approximately 0.06. The method proposed in this study produced the highest results, but relative model performance remained consistent across all studies, with Random Forest outperforming every other model. Nonetheless, future research should improve upon this methodology and include the spatial relationship of nearby pixel values.

Additionally, the features collected for this data contained only a snapshot of the temporal period covered. For example, it is not believed a terrorist attack in 1970 could have foreseen the land use of Europe in 2020, and it is unlikely the classification would have remained across the fifty years. Greater control over the spatial-temporal aspect of terrorism should be a consideration of future research.

Lastly, despite these studies' results suggesting traditional classifiers outperform neural network architectures, there remain significant areas in which deep learning could improve upon the limitations of this work. This study approached the topic from a vector-based solution, but the issue could also be applied from a computer vision approach using convolutional neural networks and key point detection. Research in this area would likely improve upon the shortcomings of this work.

The Random Forest outperformed more complex models, including multi-layered neural networks. This is likely due to Random Forest's robustness in the presence of multiple issues inherent in anomaly detection studies, of which terrorism is most assuredly a spatial anomaly. Neural network performance, however, cannot be guaranteed in the presence of large data imbalances. In fact, class imbalance is perhaps one of the greatest pitfalls of fully connected networks [48].

However, the model demonstrated false negative spatial correlation and therefore may not capture all possible attack locations in Eastern Ukraine, Northern Ireland, the island of Corsica, and the central Balkans or project to those areas subject to separatist terrorism in the future. In doing so, this study quantitatively backed previous research's understanding of the geographic difference between territorial and non-territorial terrorism. Future work should improve upon this study with respect to the consideration of surrounding pixel values among the raster data sets.

From this work, we can conclude the following: terrorist target selection is not complex and can largely be inferred. Additionally, non-separatist terrorism operates with like-spatial logic but distinct from that of separatists. Parsing the spatial dimension of territorial terrorism will likely require a unique set of explanatory variables and model selection that is not captured here.

**Author Contributions:** Conceptualization, Caleb Buffa and Vasit Sagan; methodology, Caleb Buffa and Vasit Sagan; validation, Caleb Buffa; formal analysis, Caleb Buffa; resources, Vasit Sagan; data curation, Caleb Buffa and Gregory Brunner; writing—original draft preparation, Caleb Buffa, Gregory Brunner, Vasit Sagan and Zachary Phillips; writing—review and editing, Caleb Buffa, Gregory Brunner, Vasit Sagan and Zachary Phillips; visualization, Caleb Buffa, Gregory Brunner, Vasit Sagan and Zachary Phillips; supervision, Vasit Sagan, Gregory Brunner and Zachary Phillips; project

administration, Vasit Sagan, Gregory Brunner and Zachary Phillips. All authors have read and agreed to the published version of the manuscript.

**Funding:** This research received no external funding.

**Data Availability Statement:** Publicly available datasets were analyzed in this study. Data can be found here: www.worldpop.org (accessed on 11 March 2022). Other data presented in this study are openly available at https://doi.org/10.5281/zenodo.3939050 (accessed on 11 March 2022) https://www.doi.org/10.1177/0022343310378914 (accessed on 11 March 2022) and https://doi.org/10.7910/DVN/28075 (accessed on 11 March 2022), reference numbers [27,29,30].

**Conflicts of Interest:** The authors declare no conflict of interest.

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
