# Peer review of "Predicting Terrorism in Europe with Remote Sensing, Spatial Statistics, and Machine Learning"

_ijgi, doi:10.3390/ijgi11040211_

Round 1
Reviewer 1 Report
General concept comments
The paper presents a way to develop a model to predict a terrorist attack. The determinants are the values of data obtained from publicly available sources. With the help of Five machine learning models, binary division was made into areas in which there were or were not terrorist attacks. A Random Forest model was created to predict potential attack objects. The results were validated by robust F1 and average precision scores, 0.96 and 0.97 respectively It is a very interesting idea, there are many elements that, as the authors themselves note, can be implemented into the model to make it even more credible and true. A very interesting approach is the presentation of threats based on elements of a grid measuring about 5 km, which allows you to focus not only on a given city, but even on a part of its district … Moreover, statistical analysis showed a difference in the places of attacks carried out by separatists and other groups terrorist
Article: Your new method uses GIS and focuses on trying to identify potential terrorist attack sites smaller than the country scale (current models are more regionally focused, with the state being the smallest cell considered).
Please change the information that in description of local terrorism, GIS has been used little. It is more than reference [3].
For first examples:
- Asmat, A.: The role of remote sensing in fighting against terrorism – a case of Pakistan, IAPRS, XXXVIII, (2010). Retrieved from: http://www.isprs.org/proceedings/XXXVIII/part7/b/pdf/32_XXXVIII-part7B.pdf.
- Joanna Bac-Bronowicz, Piotr Kowalczyk, Monika Bartlewska-Urban: Risk Reduction of a Terrorist Attack on a Critical Infrastructure Facility of LGOM Based on the Example of the Żelazny Most Tailings Storage Facility (OUOW Żelazny Most). Studia Geotechnica et Mechanica, 42(4), 376-387. https://doi.org/10.2478/sgem-2020-0004 (2020).
- Awatef Alsharif Shejaa Ali Alharith, Yasser Abdelazim Abdelmawgoud Samak: Fighting Terrorism More Effectively with the Aid of GIS: Kingdom of Saudi Arabia Case Study. American Journal of Geographic Information System 7(1), 15-31. (2017). doi: 10.5923/j.ajgis.20180701.02.
- Chandrakar, S., Thomas, A.: Detection and prevention of terrorist activities using GIS and remote sensing. International Journal of Computer Applications 1(14), DOI: 10.5120/311-478 (2010). Retrieved from: https://0x9.me/50FTi.
- Haney, J.: A geographic approach for teaching about terrorism. Journal of Geography 116(6), 250-262 (2017).
Review: The innovativeness of the described model is evidenced by the differences, among others concerning the territory of Poland. The DBSCAN vs RFFI model, where the first indicates, for example, Poland as a white, undisputed oasis of peace, and the second vice versa (Katowice, Krakow, Warsaw, Poznan, Lodz, Trojmiasto and Szczecin / Swinoujscie) - indicates critical places for the Polish economy, where the attack will have negative impact on the activities of the state (Katowice - a coal basin, a very large population, a huge number of victims in the event of an attack on mines; Krakow - Polish history, the cradle of the state, Warsaw - the capital, the center of government power, finances, the state activity management center, Command Center of the Armed Forces, Poznan and Lodz - financial and industrial facilities, large population, dense railway network, Trojmiasto - cavern gas, shaft mining, large population, ports, Szczecin / Swinoujscie - gas terminal ...).
Looking at the results (checking the operation of the model), it can be concluded that the goals set by the authors were achieved:
- development of a model predicting the possible sites of a terrorist event, using satellite images and socio-environmental data for this purpose,
- developing a method of correct interpretation of spatial statistics based on the assumed size of the grid cells by highlighting specific cells occurring in a given region, causal variables and trends not previously included in the spatial scale.
Suggestions for other features for the Random Forest model: availability of sensitive information in public sources (geoportals), critical infrastructure facilities, strategic value of the facility and its popularity in the media.
Specific comments
- In my opinion, the proposals should be strengthened:
- The Random Forest model has interesting assumptions that were not previously considered when modeling possible sites of a terrorist attack, it will be a very helpful tool for a specialist (primarily with experience) and should not be interpreted by non-specialists on their own.
- However, no program, even the most advanced and complex one, will be able to model a BLACK SWAN attack (an event that cannot be predicted in any way and the consequences are significant) and a course of action like LONE WOLF (current science cannot define one profile modus operandi - terrorist personality - "lone wolf").
- Figure 1 - No Sources.
- Figure 1, Figure 2, Figure 3 - no reference in the text.
- The quoted [16] did not use remote sensing
Reviewer 2 Report
Recommended for publication

Reviewer 3 Report
See the attached PDF.

Reviewer 4 Report
The paper “Predicting Terrorism in Europe with Remote Sensing, Spatial Statistics, and Machine Learning” aims to propose a model to predict terrorism in Europe. The novelty is the adopted scale. Attacks between 1970 and 2018 in Eastern and Western Europe (apart from Russia) have been included. Not only socio-environmental data but also remote sensed products have been used. Five different models (DNN, NN, Random Forest, Log Reg+SDG, SVM Ensemble) have been compared. The quality of the presentation is good; the question is original and could be of interest to a wide readership. According to my opinion the analysis is appropriated and is performed with the technical standards that this journal require.
In conclusion, according to my opinion, this article is ready to be published after a minor revision:
Some concerns:
- Many remotely sensed, geospatial, socio-environmental data have been used. I suggest to summarize them into a table to make the reading more easy and fluent;
- The selections of the remotely sensed products appears as random to the readers. It could be useful to justify your choice;
- How did you deal with the issue of the “different spatial resolution” which characterize the adopted data? Some comments, please.
- Check papers, there are some typos (i.e., lines 61, 84, etc.).
